# Investigation of the Specification Degradation Mechanism of CMOS Power Amplifier under Thermal Shock Test

**DOI:** 10.3390/mi13060815

**Published:** 2022-05-24

**Authors:** Shaohua Zhou, Cheng Yang, Jian Wang

**Affiliations:** 1School of Microelectronics, Tianjin University, Tianjin 300072, China; zhoushaohua@tju.edu.cn (S.Z.); ych2041@tju.edu.cn (C.Y.); 2Qingdao Institute for Ocean Technology, Tianjin University, Qingdao 266200, China

**Keywords:** PCB, FR-4, CMOS, power amplifier, performance degradation, thermal shock

## Abstract

To investigate the critical specifications of a power amplifier (PA) under rapidly changing temperature conditions, we fabricated and tested a 0.3–1.1 GHz complementary metal–oxide–semiconductor (CMOS) PA under thermal shock tests. The results show that high- and low-temperature shocks can accelerate the degradation of critical specifications of PAs and that the critical specifications of PAs degrade with the increasing number of shocks. The main reason for the degradation of critical specifications of PAs with increasing thermal shock tests is the mismatch of thermal expansion coefficients of printed circuit boards (PCB) with FR-4 as a substrate. The results of this paper can provide a reference for the development of temperature-robust PAs design guidelines for the implementation of temperature-robust PAs using low-cost silicon technology.

## 1. Introduction

The power amplifier (PA) is one of the core components of communication systems, which are the focus of study in the field of wireless broadband communication [1,2,3]. PAs are often used in harsh environments [4]. The characteristics of the PA have a significant temperature dependence [5]. When the temperature changes, so does the PA’s performance, which in severe cases, may cause the system to fail to perform its intended function [6]. According to the literature, 31% of electronic equipment failures at U.S. coastal bases are caused by high temperatures and humidity [7,8,9]. Likewise, a change in the specification of a PA will cause the system in which it is located to fail [10]. Therefore, there is great interest in stabilizing the performance of a PA more quickly and efficiently so that it can operate properly in rugged environments [11]. Therefore, it is necessary to understand the temperature characteristics of a PA, mainly to stabilize the performance of PAs faster and more effectively.

The temperature characteristics of PAs are a hot topic of research. For example, the temperature characteristics of PAs at specific typical temperature points, such as −5 °C [12], 25 °C [13], 125 °C [9], etc., or the temperature characteristics of PAs in specific typical temperature ranges, such as −20 °C to 120 °C [14], −40 °C to 125 °C [10], 10 °C to 100 °C [15], etc., can be investigated. These studies generally reflect the properties of PAs at a specific typical temperature or in a typical temperature range, which are gradually changed throughout the study.

However, PAs sometimes must work in environments with rapid temperature changes. These rapid temperature changes will significantly impact the specification of PAs [14]. Therefore, it becomes an urgent problem to know the pattern and mechanism of the specification degradation of PAs under the working environment of rapid temperature changes [16].

High- and low-temperature shock experiments can better characterize the properties temperature of PA under rapid changes [17]. In this paper, high- and low-temperature shock experiments were carried out with a Class A PA of 0.3–1.1 GHz, and the phenomena and causes of PA specification degradation under high- and low-temperature shocks are discussed in detail.

## 2. Designed PA and the Experimental Setup

### 2.1. The Designed PA

The schematic diagram of the 0.3–1.1 GHz CMOS Class-A PA used in the thermal shock experiments is shown in Figure 1a [18]. Considering that the linearity specification of the common source amplifier structure is better than that of the Cascode structure, a combination of a common source amplifier and resistive negative feedback structure was chosen for the third stage amplifier. The resistive negative feedback structure can match the impedance of the input and output circuits of the amplifier. At the same time, the resistive negative feedback structure can improve gain flatness and reduce nonlinear distortion. Therefore, the impedance matching of the amplifier output structure can be achieved without adding additional circuit components. The second stage amplifier was designed with a common source and gate Cascode structure. The cascaded structure of the PA can significantly improve the isolation at the input/output, and it will increase the output impedance of the transistor. The first stage amplifier was also designed with a common source and gate Cascode structure. The sizes of the transistors M0~M6 of PA used in this paper are 200 μm/0.35 μm, and M7 is a 400 μm/0.35 μm-thick oxide layer NMOS device. The on-chip stage spacing direct coupling capacitors C1 and C2 are selected to be 20 pF capacitors.

### 2.2. Thermal-Shock Test

To study the effect of rapid temperature changes on PA specifications, we conducted thermal shock tests in the shock chamber (Vötsch VT^3^ 7006 S2). The logical relationship of the test system is shown in Figure 2a, and the physical deployment of the test system is shown in Figure 2b. In this chamber, the temperature regulation range is from +50 °C to +220 °C in the hot zone and from −80 °C to +70 °C in the cold zone.

As shown in Figure 3, the temperatures in the high-temperature and low-temperature regions are 125 °C and −40 °C, respectively. The PA stays in the high-temperature and low-temperature zones during each thermal shock cycle for 15 min, respectively. After five thermal shock cycles, the PA was left in the room temperature zone for 35 min so that it could be measured after thermal shock. The input power during the thermal shock cycle was always −20 dBm, which the Agilent E8257D provided. The voltage provided by the power supply during and after the completion of the shock measurement process was always 3.3 V. The input power provided by the signal source during the shock was −20 dBm. The input power provided by the signal source during the measurement process after the completion of the shock was from −10 dBm to 18 dBm. The measured frequency was 433 MHz.

## 3. Results and Discussions

### 3.1. DC Current

The degradation of the DC current under shock and typical temperature experiments is shown in Figure 4. As seen from the results of the shock experiments, the current decreased from 113.4 mA (*P_in_* = 8 dBm) to 110.7 mA after 25 cycles of thermal shock experiments. From a typical temperature experiment, the DC current of the PA decreased from 113.4 mA (*P_in_* = 8 dBm) to 91.5 mA when the temperature rose from 25 °C to 90 °C.

When the thermal shock experiment and the typical temperature experiment were completed, the two test objects were measured at room temperature. The measurement results show that the DC current of the PA could not be restored to the initial state of the PA after 25 cycles of the thermal shock experiment. However, the DC current of the PA after the typical temperature experiment can be restored to the initial state of the PA.

This indicates that the thermal shock experiment and the typical temperature experiment have different effects on the DC current of PA. It also shows that the effect of thermal shock experiments on PA is destructive and irreversible. Finally, the reasons for current degradation under two different experimental methods were analyzed.

In the saturation region, the drain current (*I_ds_*) can be written as [19]:(1)$$I_{ds}=\frac{W_{g}\mu_{n}C_{ox}}{2L_{g}}{\left(V_{gs}-V_{th}\right)}^{2}$$
where *W_g_* is the gate width, *μ_n_* is the carrier mobility, *C_ox_* is the gate oxide capacitance per unit area, *L_g_* is the gate length, *V_gs_* is the gate-to-source voltage, and *V_th_* is the threshold voltage.

According to Equation (1), only the carrier mobility and the threshold voltage are temperature-dependent in the expression of the drain current. The two cases of thermal shock experiments and typical temperature experiments are discussed and analyzed.

#### 3.1.1. Thermal Shock Experiment

The literature reported that thermal shock experiments will increase the threshold voltage [20,21]. Therefore, when the number of thermal shocks is changed, the expression for the drain current can be written as:(2)$$I_{ds-shock}=\frac{W_{g}\mu_{n}C_{ox}}{2L_{g}}{\left(V_{gs}-V_{th-shock}\right)}^{2}$$

According to Equations (1) and (2), the ratio before and after the drain current shock can be expressed as:(3)$$\frac{I_{ds}}{I_{ds-shock}}=\frac{{\left(V_{gs}-V_{th}\right)}^{2}}{{\left(V_{gs}-V_{th-shock}\right)}^{2}}$$

Furthermore, we can obtain the extraction of the root of this ratio as:(4)$$\sqrt{\frac{I_{ds}}{I_{ds-shock}}}=\frac{\left(V_{gs}-V_{th}\right)}{\left(V_{gs}-V_{th-shock}\right)}=\frac{\left(V_{gs}-V_{th}\right)}{\left(V_{gs}-V_{th}-\Delta V_{th}\right)}$$
where:(5)$$\Delta V_{th}=(V_{gs}-V_{th})(\sqrt{\frac{I_{ds}}{I_{ds-shock}}}-1)/\sqrt{\frac{I_{ds}}{I_{ds-shock}}}$$

Additionally, the drain current of class-A PA can be expressed as:(6)$$I_{ds}=I_{DC}+i_{rf}\sin w_{0}t$$

According to Equation (6), the change in drain current can be considered when the change in DC current so that it can be obtained:(7)$$\Delta V_{th}=(V_{gs}-V_{th})\times 9.9\times 10^{-3}$$

In this way, the increment of the *V_th_* can be obtained. Therefore, the threshold voltage can also be quantitatively calculated. Through the above analysis, the main cause of current degradation due to high- and low-temperature shocks is the increase in *V_th_*, which is irreversible.

#### 3.1.2. Typical Temperature Experiment

In typical temperature experiments, the leading cause of drain current degradation is a decreased carrier mobility with increasing temperature. The expression for carrier mobility is [22]:(8)$$\mu_{n}(T)=\mu_{n}(T_{0}){\left(\frac{T}{T_{0}}\right)}^{-m}$$
where *T*_0_ = 300 K, *m* = 1.5~2.

According to Equations (1) and (8), the relationship between drain current and temperature is:(9)$$I_{ds}(T)=\mu_{n}(T_{0}){\left(\frac{T}{T_{0}}\right)}^{-m}\frac{W_{g}C_{ox}}{2L_{g}}{\left(V_{gs}-V_{th}\right)}^{2}$$

The current degrades from 99 mA to 75.8 mA (*P_in_* = 2 dBm), which is obtained according to Equation (9), the ratio of the drain current at room temperature *T_R_* (identified as *I_ds_*(*T_R_*)) and temperature *T* (determined as *I_ds_*(*T*)) can be expressed as:(10)$$\frac{I_{ds}(T_{R})}{I_{ds}(T)}=\frac{\mu_{n}(T_{0}){\left(\frac{T_{R}}{T_{0}}\right)}^{-m}}{\mu_{n}(T_{0}){\left(\frac{T}{T_{0}}\right)}^{-m}}=\frac{{\left(\frac{T_{R}}{T_{0}}\right)}^{-m}}{{\left(\frac{T}{T_{0}}\right)}^{-m}}=\frac{{\left(\frac{298}{300}\right)}^{-m}}{{\left(\frac{363}{300}\right)}^{-m}}=\frac{99}{75.8}\approx 1.31$$

Therefore, we can define *m* as 1.5.

Furthermore, the variation in carrier mobility with temperature, according to Equation (11), is known quantitatively as:(11)$$\mu_{n}(T)=\mu_{n}(T_{0}){\left(\frac{T}{T_{0}}\right)}^{-1.5}$$

The degradation phenomena and mechanisms of currents under high- and low-temperature shock experiments and typical temperature experiments are different. The main cause of degradation under thermal shock experiments is the threshold voltage increase, and this degradation is irreversible. On the other hand, the degradation under typical temperature experiments is mainly due to decreased carrier mobility with temperature increases. Nonetheless, this degradation is reversible, and the current will return to the initial state after the temperature returns.

### 3.2. Output Power

Figure 5 shows the variation in the output power of the PA under high- and low-temperature shocks (frequency = 433 MHz). A significant thermal cycling effect is observed at all input powers. For example, the output power is 19.35 dBm (*P_in_* = 10 dBm). After 25 shocks, its output power becomes 19.03 dBm, i.e., a decrease of 0.32 dBm. After 105 shocks, the output power decreases to 18.11 dBm.

The cause of output power degradation is the mismatch of the thermal expansion coefficient of the PCB material. The mismatch in the coefficient of thermal expansion of the material will cause an increase in the resistance of the solder joint, which makes the thermal loss of the resistance increase [21,23]. This means that, for a given PA input power, more energy is converted into heat in the PA. Then, the *P_out_* will decrease for a given input power as the Joule heat in the solder joint increases. The *P_out_* decreases as the thermal shock cycle increases.

### 3.3. Gain

Figure 6 shows the gain characteristics under different shocks. As shown in Figure 6, the measured gain of PA decreases with the thermal cycle. For example, the gain of the PA decreases by 1.51 dB and 0.85 dB after 105 thermal cycles for *P_in_* = −4 dBm and *P_in_* = 15 dBm, respectively.

The degradation of PA gain is similarly related to the mismatch of the thermal expansion coefficient of the PCB sheet material. A specific analysis is as follows.

The expression of gain is calculated by:(12)$$\mathrm{Gain}=P_{out}-P_{in}$$
where *P_out_* is the output power, and *P_in_* is the input power.

From Section 3.2, the output power decreases with increasing high- and low-temperature shocks. According to Equation (12), The gain will decrease as the output power decreases. The gain degrades with the increase in high- and low-temperature shocks.

### 3.4. Power-Added Efficiency

Figure 7 shows the relationship between power added efficiency (PAE) and the number of high- and low-temperature shocks at 433 MHz. PAE decreases with the increase in high- and low-temperature impact times. For example, the PAE is 20.7% (*P_in_* = 8 dBm). After 25 shocks, its PAE becomes 19.4%, indicating a decrease of 1.3% in PAE. After 105 shocks, the PAE decrease to 15.1%. When the input power is 16 dBm, the PAE decreases by 1.2% and 2.5% after 25 and 105 thermal cycles.

The following shows why PAE decreases with an increasing number of shocks. The expression of PAE is as follows [19]:(13)$$\mathrm{PAE}=\frac{P_{out}-P_{in}}{P_{DC}}\times 100\%$$
where *P_out_* is the output power, *P_in_* is the input power, and *P_DC_* is the DC power.

According to Equation (13), PAE decreases as output power decreases. Furthermore, from Section 3.2, the *P_out_* decreases with increasing high- and low-temperature shocks. Therefore, the PAE will also reduce with increasing high- and low-temperature shocks. The mismatch in the thermal expansion coefficient of the PCB material will cause the PAE to decrease.

## 4. Discussion and Conclusions

This paper takes a CMOS PA as an example to study the characteristics of the PA under high- and low-temperature shocks. The results of the experimental study showed that the critical specification of the PA degrades with the increasing number of high- and low-temperature shocks. From the previous analysis, we can observe that the main reason for the decrease in drain current is the increase in threshold voltage. Additionally, the main reason for the decrease in output power, gain and PAE is the thermal mismatch of PCB material. The change in the threshold voltage affects the transistor’s performance and changes the operating state of the transistor. Thus, it affects the performance of the PA. The thermal expansion coefficient mismatch leads to increased resistance at the solder joints. The increase in resistance at the solder joints will increase the overall thermal power consumption of the circuit. This leads to a decrease in the output power of the PA. These two conditions affect the different specifications of the PA.

This experimental method used in this paper is different from the temperature experiments mentioned in the literature [9,10,12,14,15]. The experiments in the literature (refs [9,10,12,14,15]) are ordinary temperature experiments, which are reversible. After the temperature returns to room temperature, the circuit’s performance also returns to its initial value. In contrast, the shock experiment between two temperatures conducted in this paper is destructive and irreversible. After the shock, the circuit’s performance does not return to its initial value when the temperature returns to room temperature. To better illustrate this point, the degradation phenomena and mechanisms of PA critical specifications under thermal shock experiments and typical temperature experiments are specifically discussed and quantitatively analyzed in this paper, using the DC current of PA as an example. The experimental results show that the degree of influence and reason for the thermal shock test and temperature test on the degradation of PA’s specifications are different.

High- and low-temperature shock tests can accelerate the degradation of critical specifications of PA, which is of great value and significance for the subsequent study of the lifetime of PA critical specifications. Furthermore, the thermal shock experimental method adopted in this paper can be transferred (or even directly replicated) to the degradation studies of the specifications of other electronic devices.

## Figures and Tables

**Figure 1 micromachines-13-00815-f001:**
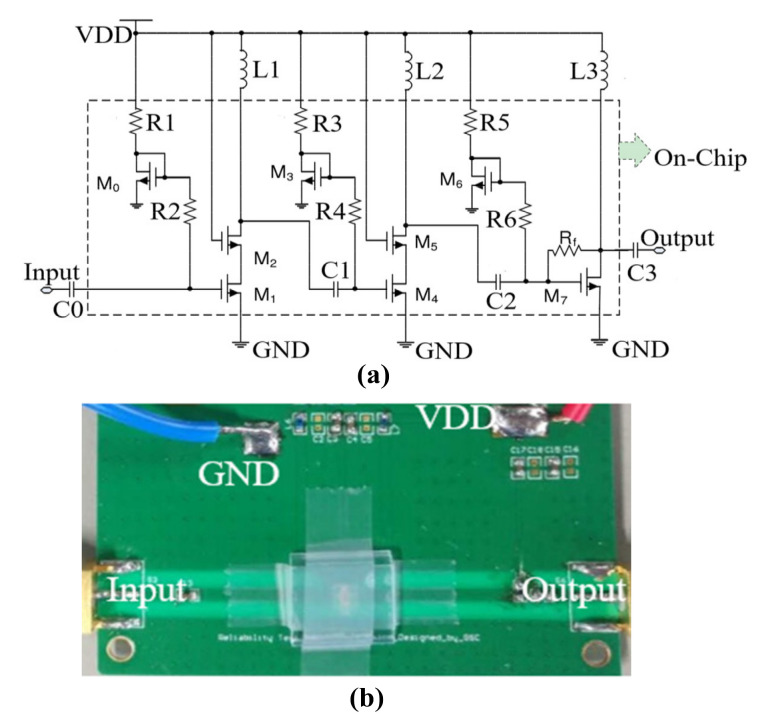
The 0.3–1.1 GHz CMOS PA: (**a**) The detailed schematic diagram; (**b**) the physical photograph.

**Figure 2 micromachines-13-00815-f002:**
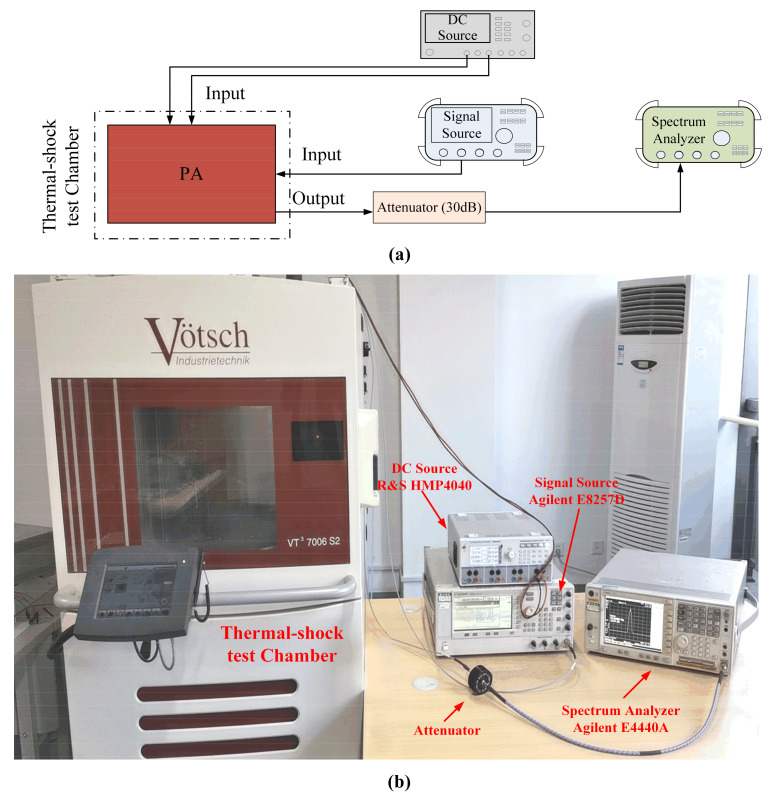
The thermal shock test environment. (**a**) The logical relationship of the test system; (**b**) the physical deployment of the test system.

**Figure 3 micromachines-13-00815-f003:**
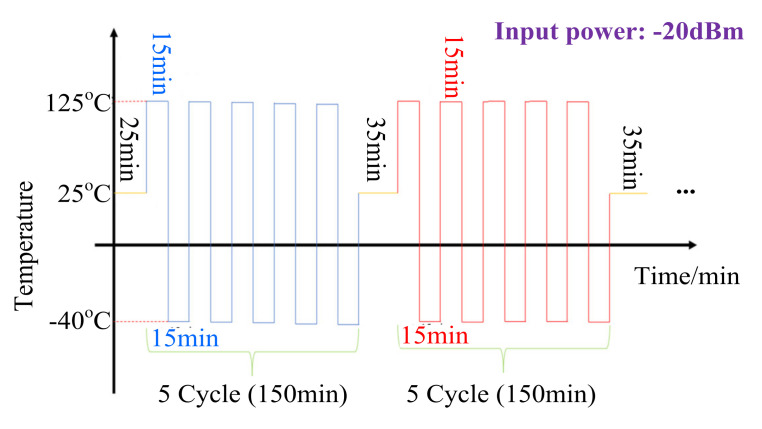
Schematic representation of the used thermal-cycling profile.

**Figure 4 micromachines-13-00815-f004:**
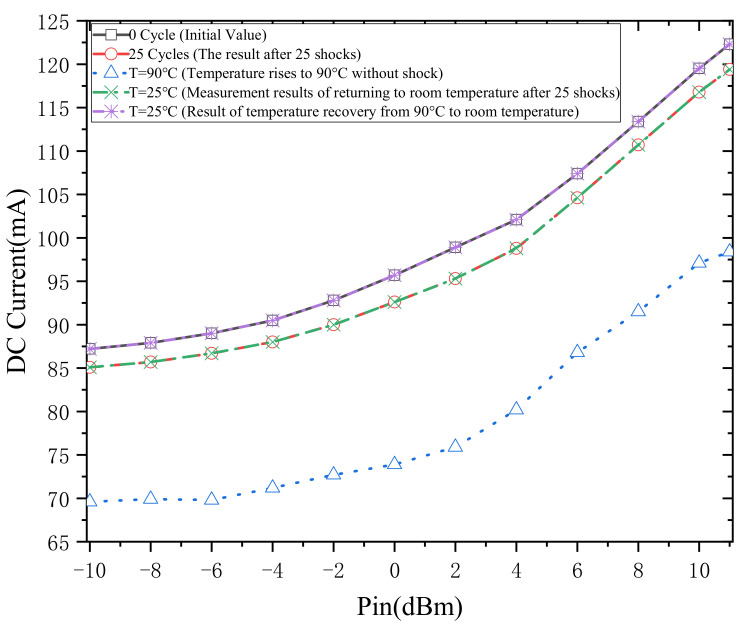
DC current of PA under thermal shock test and temperature test.

**Figure 5 micromachines-13-00815-f005:**
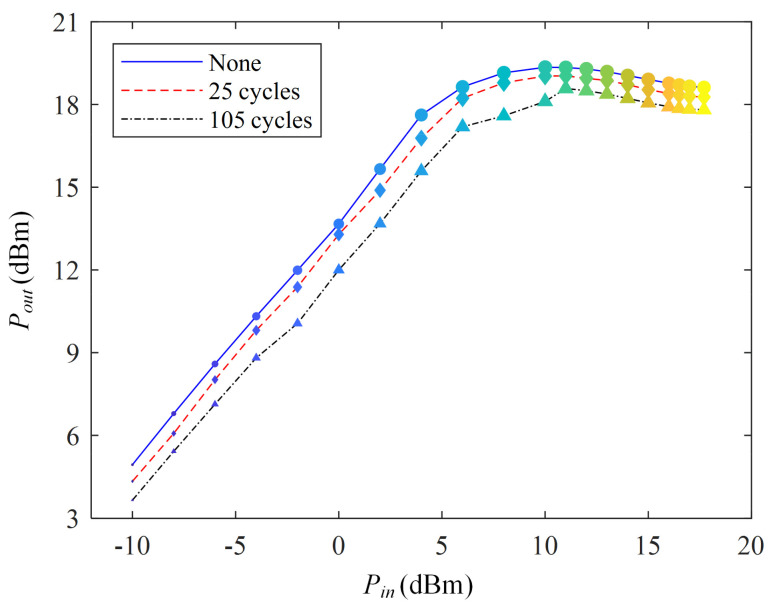
The output power of PA under thermal shock test.

**Figure 6 micromachines-13-00815-f006:**
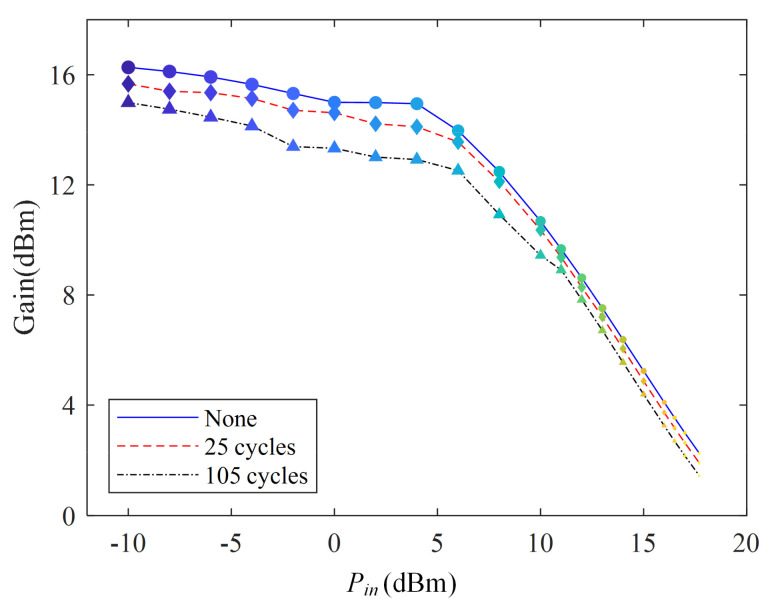
Gain of PA under thermal shock test.

**Figure 7 micromachines-13-00815-f007:**
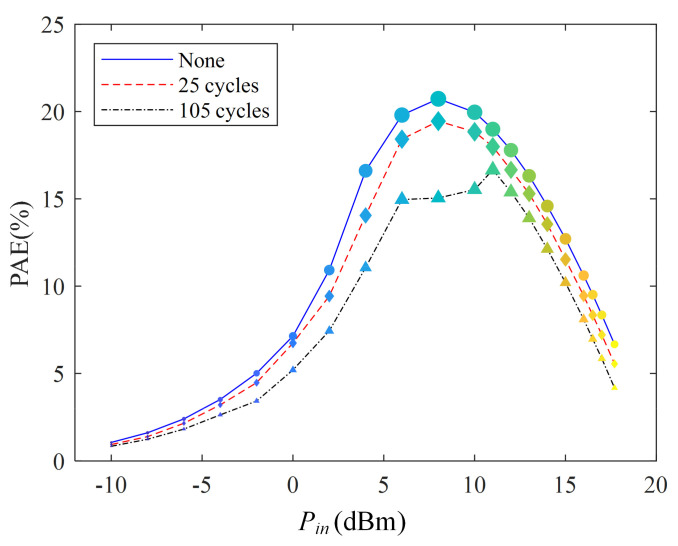
PAE of PA under thermal shock test.

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
