# Peer review of "Investigation of the Specification Degradation Mechanism of CMOS Power Amplifier under Thermal Shock Test"

_micromachines, 2022, doi:10.3390/mi13060815_

Round 1
Reviewer 1 Report
This manuscript investigates the critical specifications of the power amplifier (PA) under rapidly changing temperature conditions. The results of this work provide a reference for the development of temperature-robust PAs design guidelines. Minor comments here: under the given conditions, it would be good to include consideration of whether or not to recover to the original performance after the impact and the time it takes to recover.
Author Response
Thank you very much for your advice! Because the thermal shock experiment is a kind of irreversible experiment. It is a permanent destructive experiment, so the performance of PA cannot be restored to its original performance after the impact. We have illustrated in the manuscript the drain current example (e.g., Figure 4). For a clearer explanation, we revised these sentences “The degradation of the DC current under shock and typical temperature experiments are shown in Fig. 4. From the results of the shock experiments, the current dropped from 113.4mA (Pin=8dBm) to 110.7mA after 25 cycles of thermal shock experiments. From a typical temperature experiment, the DC current of the PA drops from 113.4mA (Pin=8dBm) to 91.5mA when the temperature rises from 25°C to 90°C. When the thermal shock experiment and the typical temperature experiment were completed, the two test objects were measured at room temperature. The measurement results showed that the DC current of the PA could not be restored to the initial state of the PA after 25 cycles of the thermal shock experiment. However, the DC current of the PA after the typical temperature experiment can be restored to the initial state of the PA.”

Reviewer 2 Report
In this paper by Shaohua Zhou, Cheng Yang and Jian Wang entitled “Investigation of the specification degradation mechanism of CMOS power amplifier under thermal shock test” the authors fabricated and tested a 0.3-1.1 GHz CMOS Power Amplifier (PA) under thermal shock tests to investigate the critical specifications of the power amplifier (PA) under rapidly changing temperature conditions. The results show that high and low-temperature shocks can accelerate the degradation of critical specifications of PAs and that the critical specifications of PAs degrade with the increasing number of shocks. The main reason for the degradation of critical specifications of PAs with increasing thermal shock tests is the mismatch of thermal expansion coefficients of Printed Circuit boards (PCB) with FR-4 as substrate.
To investigate the critical specifications of the power amplifier (PA) under rapidly changing temperature conditions, we fabricated and tested a 0.3-1.1 GHz Complementary Metal Oxide Semiconductor (CMOS) PA under thermal shock tests. The results show that high and low-temperature shocks can accelerate the degradation of critical specifications of PAs and that the critical specifications of PAs degrade with the increasing number of shocks. The main reason for the degradation of critical specifications of PAs with increasing thermal shock tests is specified as the mismatch of thermal expansion coefficients of Printed Circuit boards (PCB) with FR-4 as substrate. The results of this work seems to provide a reference for the development of temperature-robust PAs design guidelines for the implementation of temperature-robust PAs using low-cost silicon technology.
The paper is organized as follows, with the following Section titles and sub-titles::
1. Introduction
2. Designed PA and the Experimental Setup (2.1. The Designed PA, 2.2. Thermal-shock Test)
3. Results and Discussions (3.1.DC current, 3.2.Output power,3.3.Gain, 3.4.Power-added efficiency)
4.Discussion and Conclusions
The reference list containes 24 references from the years 1992 to 2022.
The schematic diagram of the 0.3-1.1 GHz CMOS Class-A PA used in the thermal shock experiments is shown in Figure 1(a).
To study the effect of rapid temperature changes on PA specifications, the authors have conducted thermal shock tests in the shock chamber (Vötsch VT3 7006 S2). The logical relationship of the test system is shown in Figure 2 (a), and the physical deployment of the test system is shown in Figure 2 (b).
The degradation of the DC current under shock and typical temperature experiments is shown in Fig. 4.
Figure 5 shows the variation of the output power of the PA under high and low temperature shocks (Frequency=433MHz).
Figure 6 shows the gain characteristics under different shocks
Figure 7 shows the relationship between power added efficiency (PAE) and the number of high and low-temperature shocks at 433MHz.
It is concluded that this paper takes a CMOS PA as an example to study the characteristics of the PA under high and low-temperature shocks. The results of the experimental study showed that the critical specification of PA degrades with the increasing number of high and low-temperature shocks.
The degradation phenomena and mechanisms of PA critical specifications under thermal shock experiments and typical temperature experiments are specifically discussed and quantitatively analyzed in the paper, using the DC current of PA as an example. The experimental results show that the influence degree and reason of thermal shock test and temperature test on the degradation of PA’s specifications are different.
A detailed good and useful work reflecting new results in the related field obtained by performing experiments. I think the paper can be considered for publication as is.
Author Response
Many thanks for your encouragement. We will do our best to improve the manuscript to make it more suitable for publication.

Reviewer 3 Report
Please, find the comments in the attached PDF file.

Author Response
Point 1: Lines 83-84 “The input power during the thermal shock cycle is always -20 dBm, which the Agilent E8257D provides”. Can the authors provide more information on the test conditions, i.e. the power supply and the input signal (amplitude, frequency) for both the thermal shock phase and the post thermal shock measurement phase?
Response 1: Thank you very much for your comment. As you suggested, we have added this section to the manuscript. The details are in Lines 84-88 as follows.
The voltage provided by the power supply during and after the completion of the shock measurement process is always 3.3V. The input power provided by the signal source during the shock is -20 dBm. The input power provided by the signal source during the measurement process after the completion of the shock is -10dBm to 18dBm. The measured frequency is 433 MHz.
Point 2: Lines 90-93 “The current degrades from 113.4mA (Pin=8dBm) to 110.7mA after undergoing 25 cycles of thermal shock experiments at temperatures ranging from -40°C to 125°C. And when the temperature increases from 25°C to 90°C, the DC current of PA degrades from 113.4mA (Pin=8dBm) to 91.5mA”. This sentence is not very clear. How is it possible the degradation is stronger for temperature variations from 25°C to 90°C than from -40°C to 125°C? Is this referred to two different currents? Please, make it clearer. Moreover, the legend of Fig. 4 is not very clear. What the cases 25 cycles (circle red) and T=90°C (triangle blue) refers to? Please, make it clearer in the text or in the figure caption.
Response 2: Thank you very much for your comment. We are sorry we did not make that clear in the original manuscript. Your understanding is correct. These are the currents in two different cases. One is the result of the thermal shock experiment, and the other is the ordinary temperature experiment. The 25 cycles (circle red) represent the measurement at room temperature after 25 thermal shocks. T=90°C (triangle blue) represents the measurement when the temperature is increased from room temperature to 90°C without a temperature shock. When the temperature returns from 90°C to room temperature. The current value will also return to its original value (black box). To be clear, we have elaborated on this point in the new manuscript, as you suggested. The details are Lines 94-97 as follows.
“From the results of the shock experiments, the current dropped from 113.4mA (Pin=8dBm) to 110.7mA after 25 cycles of thermal shock experiments. From a typical temperature experiment, the DC current of the PA drops from 113.4mA (Pin=8dBm) to 91.5mA when the temperature rises from 25°C to 90°C.”
And we have revised Figure 4.
Point 3: I do not understand well how the results of equation 10 are obtained. In particular this step:
The value m=1.5 is determined from this equation, but how 99/75.8 is obtained?
Response 3: Thank you very much for your comments. We are sorry we did not make that clear in the original manuscript. To be explained more clearly, we have revised this part.
Therefore, the value of m can be obtained according to the above formula.
Point 4: In the paper the authors state that the PA performance degradation is due to the mismatch in the coefficient of thermal expansion of the PCB with FR-4 as substrate. However, in section 3.1.1 it is shown how the temperature shock test induces permanent changes in the transistor current due to variations of the threshold voltage. What is the effect of this threshold voltage variation and how compares to the degradation due to the mismatch in the coefficient of thermal expansion of the PCB?
Response 4: Thank you very much for your comment. We are sorry we did not make that clear in the original manuscript. Based on equation (1) as , an increase in the threshold voltage leads to a decrease in the drain current. The change in threshold voltage affects the transistor's performance and makes the operating state of the transistor change. Thus, it affects the performance of the PA. And the PCB material thermal expansion coefficient mismatch affects the power consumption of the PA as a whole. The thermal expansion coefficient mismatch leads to increased resistance at the solder joints. The increase in resistance at the solder joints will increase the overall thermal power consumption of the circuit. This leads to a decrease in the output power of the PA. These two conditions affect the different specifications of the PA. A severe mismatch in the thermal expansion coefficient of the PCB will lead to the detachment of the solder joint from the PCB, failing the PA. The details are Lines 201-209 as follows.
“From the previous analysis, we can know that the main reason for the decrease of the drain current is the increase of the threshold voltage. And the main reason for the decrease of output power, gain and PAE is the thermal mismatch of PCB material. The change in threshold voltage affects the transistor's performance and makes the operating state of the transistor change. Thus, it affects the performance of the PA. The thermal expansion coefficient mismatch leads to increased resistance at the solder joints. The increase in resistance at the solder joints will increase the overall thermal power consumption of the circuit. This leads to a decrease in the output power of the PA. These two conditions affect the different specifications of the PA.”
Point 5: In the introduction it is reported that results on PA degradation at fixed temperature are reported in literature (ref 9, 10, 12, 14, 15). How the results presented in these references compare to the results obtained by the authors (thermal shock tests between two different temperatures)?
Response 5: Thank you very much for your suggestion. The experiments in the literature (ref 9, 10, 12, 14, 15) are ordinary temperature experiments, which are reversible. After the temperature returns to room temperature, the circuit's performance also returns to its initial value. In contrast, the shock experiment between two temperatures conducted in this paper is destructive and irreversible. After the shock, the circuit's performance does not return to its initial value when the temperature returns to room temperature. This point we have illustrated in the manuscript with the example of the drain current (e.g., Figure 4), which shows the difference between the two experiments. The experiments conducted in this manuscript can be used in the future for accelerated degradation of electronic components, for rapid knowledge of changes in specifications, and for predicting the lifetime of electronic components. The details are Lines 210-220 as follows.
“This experimental method used in this paper is different from the temperature ex-periments mentioned in the literature [9,10,12,14,15]. The experiments in the literature (ref 9, 10, 12, 14, 15) are ordinary temperature experiments, which are reversible. After the temperature returns to room temperature, the circuit's performance also returns to its ini-tial value. In contrast, the shock experiment between two temperatures conducted in this paper is destructive and irreversible. After the shock, the circuit's performance does not return to its initial value when the temperature returns to room temperature. To better il-lustrate this point, the degradation phenomena and mechanisms of PA critical specifica-tions under thermal shock experiments and typical temperature experiments are specifi-cally discussed and quantitatively analyzed in this paper, using the DC current of PA as an example.”
Point 6: Please, check the manuscript for errors and typos: for example, at line 107 “Vgs is the gate voltage” should be “Vgs is the gate-to-source voltage”; at line 158 out in Pout must be subscript.
Response 6: Thank you very much for your suggestion. We have followed your recommendations and made changes to the new manuscript. The details are as follows:
- Line 111: Vgs is the gate-to-source voltage
Line 163: Then, the Pout will decrease for given input power as the Joule heat in the solder joint increases.

Round 2
Reviewer 3 Report
The authors have revised the paper according to the Reviewers' comments. I think it is suitable for publication.